# Frailty Scales for Prognosis Assessment of Older Adult Patients after Acute Myocardial Infarction

**DOI:** 10.3390/jcm10184278

**Published:** 2021-09-21

**Authors:** Sergio García-Blas, Clara Bonanad, Agustín Fernández-Cisnal, Clara Sastre-Arbona, Maria-Arantzazu Ruescas-Nicolau, Jessika González D’Gregorio, Ernesto Valero, Gema Miñana, Patricia Palau, Francisco J. Tarazona-Santabalbina, Vicente Ruiz Ros, Julio Núñez, Juan Sanchis

**Affiliations:** 1Cardiology Department, University Clinic Hospital of Valencia, 46010 Valencia, Spain; sergiogarciablas@gmail.com (S.G.-B.); clarabonanad@gmail.com (C.B.); fecia82@gmail.com (A.F.-C.); clarasastre90@hotmail.com (C.S.-A.); jessikabeg89@gmail.com (J.G.D.); ernestovaleropicher@hotmail.com (E.V.); gemineta@gmail.com (G.M.); patripalau@gmail.com (P.P.); vicente.ruiz@uv.es (V.R.R.); yulnunez@gmail.com (J.N.); 2Instituto de Investigación Sanitaria INCLIVA, 46010 Valencia, Spain; 3Centro de Investigación Biomédica en Red de Enfermedades Cardiovasculares (CIBERCV), 28029 Madrid, Spain; 4Department of Medicine, Faculty of Medicine, University of Valencia, 46010 Valencia, Spain; 5Faculty of Physiotherapy, Universitat de Valencia, 46010 Valencia, Spain; Arancha.Ruescas@uv.es; 6Geriatrics Department, Hospital Universitario de la Ribera, 46600 Alzira, Spain; fjtarazonas@gmail.com

**Keywords:** frailty, acute myocardial infarction, Fried’s frailty score, Clinical Frailty Scale

## Abstract

We aimed to compare the prognostic value of two different measures, the Fried’s Frailty Scale (FFS) and the Clinical Frailty Scale (CFS), following myocardial infarction (MI). We included 150 patients ≥ 70 years admitted from AMI. Frailty was evaluated on the day before discharge. The primary endpoint was number of days alive and out of hospital (DAOH) during the first 800 days. Secondary endpoints were mortality and a composite of mortality and reinfarction. Frailty was diagnosed in 58% and 34% of patients using the FFS and CFS scales, respectively. During the first 800 days 34 deaths and 137 admissions occurred. The number of DAOH decreased significantly with increasing scores of both FFS (*p* < 0.001) and CFS (*p* = 0.049). In multivariate analysis, only the highest scores (FFS = 5, CFS ≥ 6) were independently associated with fewer DAOH. At a median follow-up of 946 days, frailty assessed both by FFS and CFS was independently associated with death and MI (HR = 2.70 95%CI = 1.32–5.51 *p* = 0.001; HR = 2.01 95%CI = 1.1–3.66 *p* = 0.023, respectively), whereas all-cause mortality was only associated with FFS (HR = 1.51 95%CI = 1.08–2.10 *p* = 0.015). Frailty by FFS or CFS is independently associated with shorter number DAOH post-MI. Likewise, frailty assessed by either scale is associated with a higher rate of death and reinfarction, whereas FFS outperforms CFS for mortality prediction.

## 1. Introduction

Frailty is a syndrome characterized by reduced physiological reserves, which increases vulnerability to stressors and is associated with disability, morbidity and mortality [1,2]. Its prevalence in European countries ranges from 4 to 21% (25–50% in >85 years), and it is commonly associated with cardiovascular disease [3]. This syndrome is frequently present among older people with acute coronary syndrome (ACS); in an unselected cohort of patients with ACS aged ≥ 80 years 27.3% were classified as frail and 38.5% pre-frail [4]. Frail patients with ACS have more complex coronary artery disease, longer hospital stays, and are at higher risk of events at follow-up (mortality, myocardial infarction and bleeding); moreover, they are less likely to receive evidence-based therapies or invasive strategies [5,6,7,8,9,10,11].

Challenges to implementing frailty assessment in daily clinical practice include the wide range of different scales available. European Society of Cardiology clinical practice guidelines recommend frailty evaluation in patients with acute coronary syndrome for risk stratification and therapeutic decisions, and include a list of 21 outcome instruments to measure frailty [12]. There are two kinds of approaches to evaluate frailty: the first considers frailty as a phenotype of poor physical function and is based mainly on two objective measures (grip strength and gait speed); the Fried’s Frailty score (FFS) is the most widely used example of this model [13]. The second considers frailty as the consequence of accumulated deficits such as comorbidities and disabilities, identified from symptoms and laboratory data. Within this approach, the Clinical Frailty Scale (CFS) is a screening instrument based on the subjective clinical judgment of the healthcare professional which classifies patients into one of nine categories, from fit to extremely frail [14]. Both scales have proved useful for prognosis assessment after ACS [8,9,15,16]. However, data comparing these scales or their combination are scarce. 

## 2. Material and Methods

### 2.1. Study Population

The present work is a substudy of the randomized trial “Intervention in Frailty Versus Usual Care in Frail Patients After an Acute Myocardial Infarction” (ClinicalTrials.gov NCT02715453). Full details are published elsewhere [17]. In brief, we conducted a single center randomized trial in older adult patients with pre-frailty or frailty hospitalized for acute myocardial infarction. Inclusion criteria included acute myocardial infarction (with or without ST-segment elevation); age ≥70 years, and pre-frailty or frailty according to FFS (1–2 points or ≥3 points, respectively). Patients were excluded if they had severe concomitant disease that would preclude participation in the study or cognitive impairment (>3 mistakes in the Pfeiffer test). The study was reviewed and approved by the Clinical Research Ethics Committee of the Clinic University Hospital in Valencia.

### 2.2. Variables and Definitions

Demographic, relevant clinical history and admission data were collected. Diagnostic work-up and treatment was indicated at the discretion of the attending physicians. Frailty was evaluated with both the FFS and the CFS on the day before discharge. FFS was calculated assigning 0 or 1 points (according to the FFS definitions) to the following parameters: unintentional weight loss (>4.5 kg in the preceding year), low physical activity (Minnesota Leisure Time Activity questionnaire), slowness (time to walk 15 feet), weakness (grip strength using a hand-held isometric dynamometer) and exhaustion (Center for Epidemiological Studies–Depression scale) [13]. Pre-frailty was defined as 1 or 2 and frailty as ≥3 points in total. For the CFS, each subject was assigned a score between 1–9, ranging from a fit, healthy independent individual to complete functional dependence [14]. Frailty was considered if CFS was ≥5 [14]. Both assessments were performed by trained personal with extensive relevant experience. Both assessments were performed by trained personnel. GRACE score was calculated in each patient due to its prognostic value in the setting of myocardial infarction [18]. In addition, Charlson Comorbidity Index was included in the collected variables, due to its association with a poor prognosis independently of frailty [11,19].

### 2.3. Endpoints

The primary endpoint was number of days alive and out of hospital (DAOH) during the first 800 days. This period was the longest follow-up of the last patient included and was chosen to homogenize and use an absolute number of days. 

Secondary endpoints were all-cause mortality and a composite of mortality and reinfarction. These events were recorded at the longest available follow-up for each patient, to maximize the number of events. 

Follow-up data was recorded by either reviewing patients’ electronic medical records or contacting them directly.

### 2.4. Statistical Analysis

Baseline patient characteristics are reported as mean and standard deviation for continuous variables, and as frequencies (percentages) for categorical variables. FFS and CFS were expressed as ordinal variables by percentage in each category. Correlation between the two scales was assessed by Spearman’s rank correlation coefficient. 

Shapiro-Wilk’s test indicated a non-normal distribution for the primary endpoint. Frailty scales are ordinal variables. As they have been analyzed as categorical, continuous and dichotomous in the literature [20] we used all these scale types. Correlation between frailty scales and number of DAOH was evaluated as follows: (1) When frailty scales were considered as continuous a Spearman’s rank correlation was used. Univariate generalized linear models were constructed and linear assumptions were tested and transformed, if appropriate, with fractional polynomials. (2) When considered as categorical, a Kruskall-Wallis test was performed with a post-hoc analysis comparing each one, taking the lowest value (FFS = 1, CFS = 2) with Mann-Whitney’s test as a reference. (3) Finally, frailty scores were dichotomized as supported by the literature [13,14] (FFS <3; ≥3, CFS < 5; ≥5) and compared with Mann-Whitney’s test, isolated and combined into four categories (FFS ≤ 3 and CFS ≤ 5, FFS ≥ 3 and CFS ≤ 5, FFS ≤ 3 and CFS ≥ 5, FFS ≥ 3 and CFS ≥ 5). 

Univariate analysis was performed to evaluate the association between the remaining study variables and the primary endpoint using Spearman’s rank correlation coefficient for quantitative, Kruskall-Wallis test for ordinal and Mann–Whitney U test for dichotomic. All significantly or near-significantly associated variables (*p* < 0.10) were included in multivariate analysis. A backward stepwise selection was used, with Akaike information criteria as a stopping criterion. The association of Fried and CFS with the primary endpoint was explored with multivariate generalized linear regression models (GLM). Three GLMs were constructed introducing frailty variables as categorical, continuous and dichotomous, as previously explained. 

Individual association of FFS and CFS with each secondary endpoint was explored by Cox regression. Next, univariate analysis was performed considering the clinical variables presented in Table 1. Finally, all significantly or near-significantly associated (*p* < 0.10) variables were included in multivariate Cox regression analysis using backward stepwise methodology. Results were expressed by hazard ratio or beta coefficient of each variable with 95% confidence interval and statistical significance (*p*).

## 3. Results

In total, 150 consecutive patients were included from January 2016 to August 2018; their baseline characteristics are shown in Table 1. In brief, 62% were male, mean age was 80 years (standard deviation: 5.9), 19.3% were admitted for ST segment elevation myocardial infarction, coronary angiography was performed in 95.3% and 58.7% of patients underwent revascularization. The FFS was ≥3 in 87 (58%) patients, and 51 (34%) had a CFS ≥5; distribution of patients within FFS and CFS score is shown in Figure 1. 

The two frailty scales showed significant positive correlation (Rho Spearman = 0.297, *p* < 0.001) (Figure 2).

During the first 800 days evaluated for the primary endpoint, there were 34 deaths (22.7%), and 137 admissions for any cause in 66 patients, whereas 74 (49.3%) patients remained alive and without readmissions during the 800 days. The primary endpoint varied significantly across the different scores in both FFS (*p* < 0.001) and CFS (*p* = 0.049) evaluated individually, driven mainly by the highest scores of each (Figure 3). Additionally, frailty status as defined by either scale (i.e., dichotomic variable) was associated with significantly fewer DAOH (FFS ≥ 3 *p* = 0.001, CFS ≥ 5 *p* = 0.046). Other variables associated with the primary endpoint in the univariate analysis were: diabetes mellitus, prior myocardial infarction, stroke or antiplatelet therapy, ST-segment elevation myocardial infarction at presentation, atrial fibrillation, hemoglobin, creatinine, glomerular filtration rate, and revascularization during index procedure.

Multivariate analysis showed that only the highest recorded scores on the frailty scales (FFS = 5, CFS ≥ 6), compared to their lowest values (FFS = 1, CFS = 2) were independently associated with fewer DAOH, alongside prior stroke and creatinine levels (Table 2, Figure 4). Z-values of this variables indicated that the two scales contribute similarly to the model (Figure 4).

To determine the combined value of FFS and CFS, the sample was divided into four groups: non-frail on both scales (FFS < 3 and CFS < 5); frail only by CFS (FFS < 3 but CFS ≥ 5), frail only by FFS (FFS ≥ 3 but CFS < 5) and defined by both as frail. The Kruskal-Wallis test showed differing DAOH across the subgroups, which were lowest in the last group (FFS ≥ 3 and CFS ≥ 5), as illustrated in Figure 5. However, these differences did not reach statistical significance in multivariate analysis.

Regarding secondary endpoint analysis, at a median follow-up of 946 days, 51 (34%) patients had the composite event of death or non-fatal reinfarction. Univariate analysis showed that a higher FFS was associated with mortality and reinfarction at follow-up, both evaluated as an ordinal variable and comparing non-frail (FFS 1–2) with frail (FFS ≥ 3) patients. Frailty evaluated by CFS was only associated with this composite endpoint using the prespecified threshold of 5 points. Multivariate Cox regression analysis confirmed that the FFS (ordinal and dichotomized) was independently associated with mortality and reinfarction (Table 3), together with prior myocardial infarction, antiplatelet treatment, atrial fibrillation, and glomerular filtration rate. CFS ≥ 5 remained statistically significant when added to the latter model. 

Finally, 34 (22.7%) patients had died by longest follow-up. FFS was significantly associated with mortality, both evaluated as an ordinal variable and dichotomized, whereas CFS was not significantly associated with this event. A clinical multivariate model including age, diabetes mellitus, prior stroke, atrial fibrillation, and creatinine levels at admission was built to predict mortality. After adding the FFS to the clinical model, it remained as an independent predictor of mortality (HR = 1.51 [CI 95% 1.08–2.10] *p* = 0.015), while age was no longer significantly associated.

## 4. Discussion

The main finding of our study is that frailty as assessed by FFS or CFS is independently associated with fewer DAOH during the first 800 days after acute myocardial infarction. Both scales are useful and may be chosen for this prognostic assessment. The combination of frailty defined by FFS and CFS may identify the highest risk patients, although its independent association has not been demonstrated. Additionally, frailty assessed by either of the two scales is associated with a higher rate of death and reinfarction, while the FFS outperforms CFS for mortality prediction at a median follow-up of 2.6 years. 

Widespread application of frailty assessment requires a comprehensive comparison of different scales in order to determine: (1) feasibility and performance in different clinical scenarios (i.e., acute vs. chronic setting), (2) prognostic information, (3) role in guiding therapeutic decisions. The overwhelming range of literature and instruments described discourage clinical application, yet efforts should be made to identify frailty in a simple and practical way.

The results of our study support the use of frailty scales after MI for further risk assessment. A wealth of scientific evidence agrees with this, and their use is recommended in clinical practice [12,21]. However, several obstacles to their widespread use include the plethora of different scales available. A recent systematic review identified 51 frailty assessment instruments with considerable diversity of characteristics [20]. The FFS is a well validated tool that considers frailty as a phenotype of poor physical function, and therefore relies mostly on physical performance tests (grip strength and gait speed). Being thus a time-consuming test, it is difficult to apply in an acute event, where the clinical situation may also limit mobility and the frailty of the individual can be overestimated. CFS is a judgement-based tool to stratify degree of fitness and frailty, and rather than using specific tests it summarizes the data received from anamnesis and medical history review, as each point on the scale corresponds with a description of frailty. Accordingly, it seems more feasible for the acute setting, although concerns may arise about its subjectivity, particularly when used by inexperienced personnel. Both scales have been validated in different clinical scenarios, but direct comparison of those scales after MI is lacking [22,23,24,25]. We found both to be independent predictors of DAOH, with a similar performance when included in a multivariate model. This finding leads us to two conclusions: the two can be used interchangeably, and their combination further stratifies risk. Given the above features, CFS may be preferable for use at admission and FFS at discharge or soon after, providing that physical status is adequately recovered.

Some evidence indicates that the FFS and CFS provide similar prognostic information in different clinical scenarios. In a cohort of patients > 65 years treated in the emergency department, Lewis et al. found that CFS and FFS showed a similar performance for prediction of poor post-discharge outcomes (death, poor quality of life, need for community services or readmission to ED) [26]. In a cohort of 307 patients > 65 years admitted to a geriatric ward, both FFS and CFS score significantly predicted mortality at follow-up, while unplanned readmissions for any cause were only associated with CFS [27]. Furthermore, integrating frailty assessment via either FFS or CFS criteria to traditional surgical risk scores provided additive value in identifying patients at risk of poor functional survival at one year after cardiac surgery [28]. Few studies have focused on the possible value of combining frailty scales. Results from the Cardiovascular Health Study showed that an index of cumulative deficits (including a total of 48 deficits) outperformed the FFS, but the authors suggested that combining both approaches conferred an increased precision in mortality risk discrimination [29].

Exploring the scale-related data in more depth, we found that only the highest values (i.e., the frailest patients) correlated with lower survival out of hospital, indicating that specific prevention strategies should be targeted to this subgroup. Validated thresholds for frailty (FFS ≥ 3 and CFS ≥ 5) were not independently associated with the primary endpoint. One possible explanation for this finding is the lack of fit patients in our cohort, which may have amplified the differences between frail and non-frail subgroups. However, median survival out of hospital during the first 800 days for non-frail patients in our study was 800 (interquartile range = 5), hence including fit patients would not be expected to improve these data significantly. Moreover, frailty is a dynamic process and patients with borderline scores may potentially improve their status, whereas extremely frail patients are unlikely to recover [30]. Either way, our results highlight that only extreme frailty may influence prognosis, which should be considered for therapeutic decision-making in older patients.

A novel primary endpoint was selected in this study, DAOH. This includes mortality and its timing, and, interestingly, all-cause admissions weighted by length of stay. We believe that this is a more useful approach than simply including all-cause readmissions, because the latter attaches the same importance to a short stay for a non-serious cause as to a prolonged one for a serious condition. Moreover, in the older patients, admissions impact on dependence and loss of quality of life, which are of crucial importance in this population [31]. In this line of thought, an ongoing randomized trial will use this primary endpoint (DAOH) to compare an invasive vs. conservative strategy in frail NSTEMI patients [32].

Regarding secondary endpoints, the FFS outperforms CFS for mortality and reinfarction prediction in our sample. This may reflect the fact that FFS is a more physical scale, and accordingly may identify lower physiologic reserve, which impacts directly on hard endpoints. In contrast, CFS records accumulated deficits, which entail vulnerability for readmission for any cause; this principally affects survival out of hospital. In other words, patients with a high score in CFS but not defined as frail according to FFS are prone to readmissions but potentially have sufficient physiologic reserve to survive.

Some limitations of this study must be acknowledged. First, the limited number of patients precludes drawing any firm conclusions regarding the combined FFS and CFS subgroups in multivariate analysis. Second, excluding fit patients may limit generalization of the results to the elderly population as a whole. Finally, as this sample was recruited from a cardiology ward, some selection bias may have excluded the frailest patients.

## 5. Conclusions

Frailty as evaluated by either FFS or CFS is independently associated with shorter number days alive out of hospital after acute myocardial infarction. This is mainly driven by the highest scores in their respective scales. Both scales may be used in this setting, and their combination might provide additional value for risk prediction. Additionally, frailty assessed by either scale is associated with a higher rate of death and reinfarction, whereas the FFS outperforms CFS for mortality prediction.

## Figures and Tables

**Figure 1 jcm-10-04278-f001:**
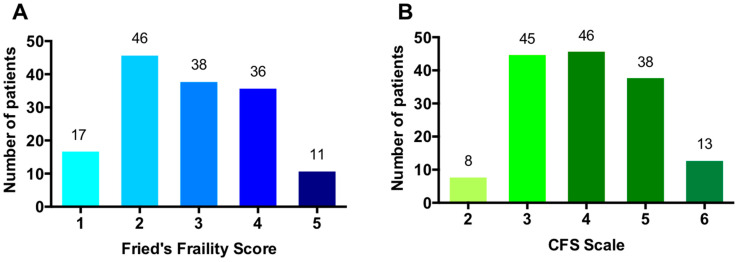
Patient distribution according to frailty scales (**A**: Fried’s Frailty Score, **B**: Clinical Frailty Scale [CFS]).

**Figure 2 jcm-10-04278-f002:**
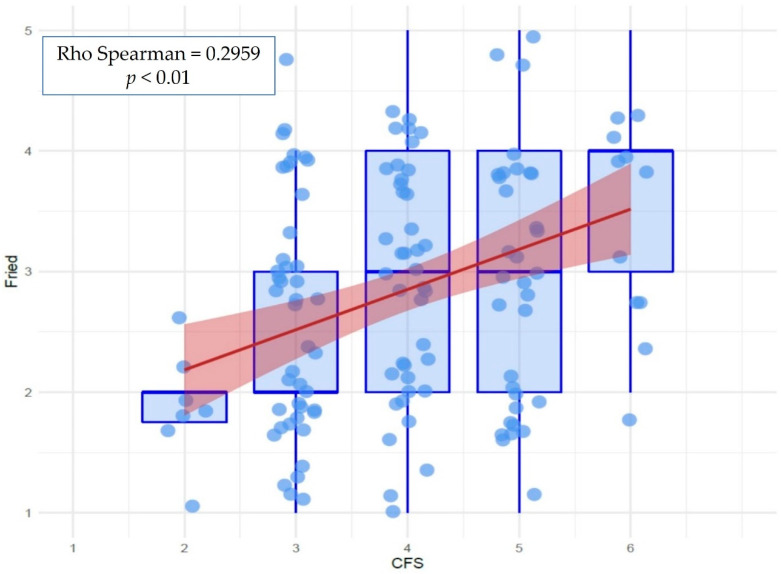
Correlation between Frailty scales. CFS: Clinical Frailty Scale.

**Figure 3 jcm-10-04278-f003:**
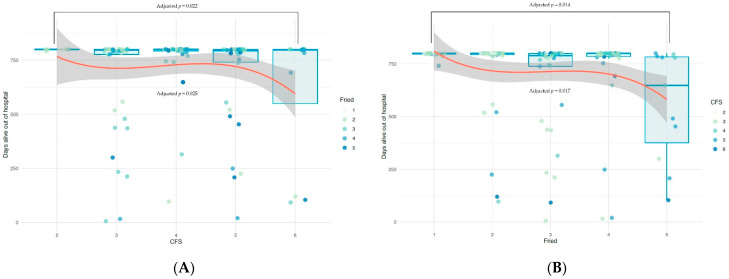
Relationship between primary endpoint (DAOH) and frailty scales ((**A**): Fried scale, (**B**): Clinical Frailty Scale [CFS]) considered as categorical (boxplots) and continuous (red line representing grade 3 factorial polynomial adjustment). P values adjusted by multivariate GLMs including the two frailty scales, relevant clinical variables and those with *p* < 0.1 in the univariate analyses.

**Figure 4 jcm-10-04278-f004:**
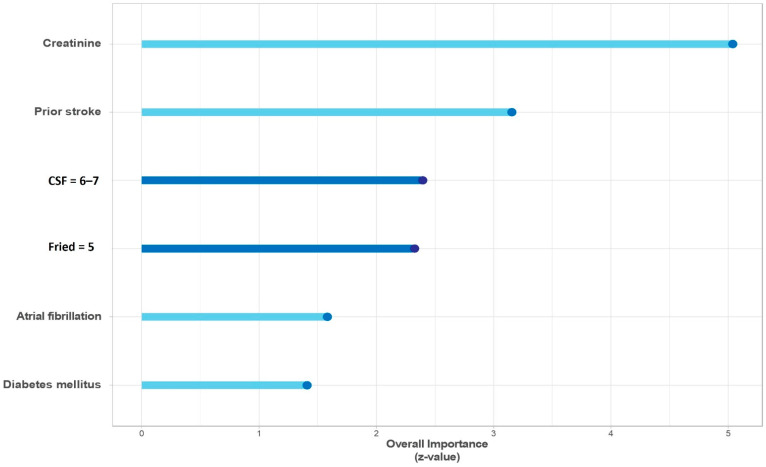
Multivariate model for primary endpoint.

**Figure 5 jcm-10-04278-f005:**
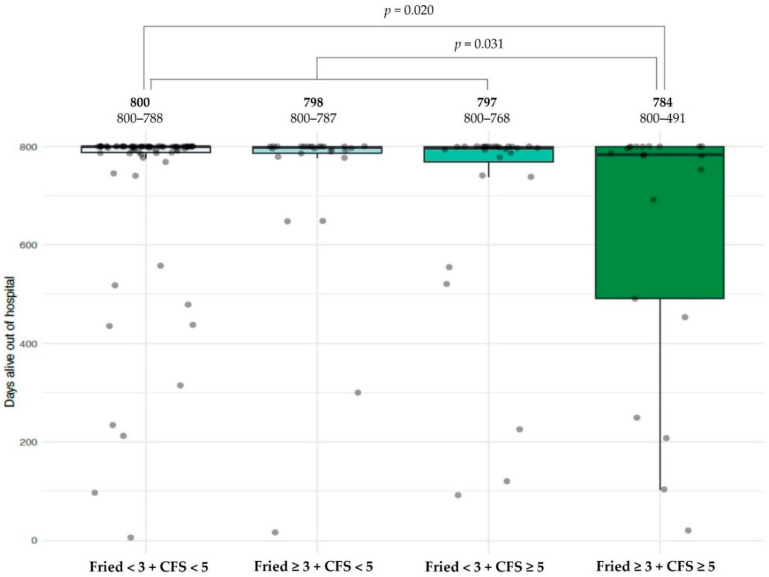
Relationship between primary endpoint (DAOH) and dichotomized (FFS ≥ 3, Clinical Frailty Scale [CFS] ≥ 5) and combined frailty scales. *p* values are related to univariate analysis.

**Table 1 jcm-10-04278-t001:** Baseline characteristics, *n* = 150. Values are expressed as mean ± standard deviation or number (percentage).

Age	80.02 ± 5.9
Male	93 (62%)
Hypertension	130 (86.7%)
DM	67 (44.7%)
Smoker	25 (16.7%)
Dyslipidemia	73 (48.7%)
Prior MI	46 (30.7%)
Prior HF admission	9 (6%)
Prior stroke	19 (12.7%)
Peripheral artery disease	15 (10%)
Chronic lung disease	26 (17.3%)
STEMI	29 (19.3%)
Hb	13.8 ± 12.1
Creatinine	1.18 ± 0.53
LVEF	54.4 ± 12.8
Coronary angiography	143 (95.3%)
PCI	84 (56%)
CABG	4 (2.7%)
GRACE score	192.6 ± 47.9
Charlson index	2.0 ± 1.8

Abbreviations: DM: diabetes mellitus; MI: myocardial infarction; HF: heart failure; STEMI: ST segment elevation myocardial infarction: Hb: Hemoglobin; LVEF: left ventricular ejection fraction; PCI: percutaneous coronary intervention; CABG: coronary artery bypass grafting.

**Table 2 jcm-10-04278-t002:** Multivariate generalized linear models for primary endpoint using frailty scales as categorical.

Characteristic.	Beta	95%CI	*p* Value
FFS ^a^			
FFS = 2	−41	−137, 55	0.401
FFS = 3	−65	−165, 35	0.203
FFS = 4	−17	−119, 85	0.748
FFS = 5	−174	−311, −36	0.014
CFS ^a^			
CFS = 3	−86	−217, 44	0.198
CFS = 4	−21	−152, 111	0.758
CFS = 5	−56	−191, 79	0.418
CFS ≥ 6	−188	−347, −29	0.022
Prior stroke	−131	−216, −45	0.003
Creatinine (mg/dL)	−133	−186, −79	<0.001

FFS: Fried’s frailty score; CFS: Clinical frailty scale; CI = Confidence Interval. ^a^ Compared with the reference category (FFS = 1, CFS = 2).

**Table 3 jcm-10-04278-t003:** Multivariate Cox regression. Impact of FFS and CFS on endpoints at follow-up.

	FFS (Points)			FFS ≥ 3 Points			CFS (Categories)			CFS ≥ 5 Category		
	HR	CI 95%	*p*	HR	CI 95%	*p*	HR	CI 95%	*p*	HR	CI 95%	*p*
Mortality and reinfarction	1.54	1.19–2.01	0.001	2.70	1.32–5.51	0.006	1.10	0.79–1.53	0.59	2.01	1.10–3.66	0.023
Mortality	1.51	1.08–2.10	0.015	1.66	0.64–4.30	0.30	0.90	0.59–1.36	0.60	0.88	0.41–1.88	0.75

CI: confidence interval; HR: hazard ratio.

## Data Availability

The data underlying this article will be shared on reasonable request to the corresponding author.

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
