# Peer review of "Frailty Scales for Prognosis Assessment of Older Adult Patients after Acute Myocardial Infarction"

_jcm, 2021, doi:10.3390/jcm10184278_

Round 1

Reviewer 1 Report

Overall, very nice study of an important subject. The statistical analysis was complicated, but it was both done and presented very well. 

Comments:   Line 81: What does "extensive relevant experience" mean? Perhaps remove this phrase and leave it as trained individuals.   Table 1: GRACE score should be GRACE risk score Charlson index should be Charlson Comorbidity Index Both should be mentioned or discussed in the Methods section as well   Have these scores been associated with frailty in the literature?   Lines 144-145: in the statement "frailty status as defined", are you referring to binary (frailty versus no frailty) or as a continuous variable?   The rationale given for non-significance when scores are combined for multivariate analysis is sample size. If truly due to limited numbers, the number in each group assignment should be described in the Figure (Figure 5) or stated in the results clearly.    I do not believe that there is sufficient data to prove that combining scales improves prognostication, one can only presume by using both together it would be better clinically per say. The data DO however strongly suggest that one or the other is sufficient for the primary endpoint and probably Fried score better overall when taking into account the secondary endpoint. Therefore, the statements in Line 195/Line 267 should be less demonstrative given the lack of this data.   Revisions:   Line 19: Change "Fried" to "Fried's Frailty Scale?" Make this change throughout and abbreviate FFS as a suggestion? Throughout this paper I have seen "Fried score," "Fried's scale," "Fried scale," "Fried phenotype," "Fried" - important to use the same phrase throughout for consistency   Line 20: What is "AMI?"  if it is acute myocardial infarction, that abbreviation introduced prior and "from" should change to "with"   Line 24-25 is confusing: why include "in 66 patients?"   Line 25-26: should clarify that DAOH decreased with increasing scores   Line 37: add "a syndrome" after "is"   Line 38: remove "in those older adults"   Line 42: Change "prefrail" to "pre-frail"   Line 45: Change "strategy" to "strategies"   Line 50: Change "approach" to "approaches"   Line 64: Change "prefailty" to "pre-frailty"   Line 67: Change "could" to "would"   Line 81: Change "personal" to "personnel"   Line 94: Change "of" to "in"   Line 131: Change "patients, while..." to "patients, and CFS was >5 in 51 (34%) of patients"   Line 144: Change "additionally" to "In addition,"   Line 193: Change "findings" to "finding" and nd "are" to "is" because only one finding is provided the way the statement is made   Line 216: Change "personal" to "personnel"   Line 225: Change "reattendance" to "readmission"   Line 227: Change "predicated" to "predicted"   Line 266: Add "in their respective scales" after "driven by the highest scores.."

Author Response

Reviewer 1

Overall, very nice study of an important subject. The statistical analysis was complicated, but it was both done and presented very well. 

Comments:  

  • Line 81: What does "extensive relevant experience" mean? Perhaps remove this phrase and leave it as trained individuals.  

We agree with the reviewer that trained individuals is more precise and not subjected to interpretations. It was changed accordingly.

“Both assessments were performed by trained personnel”

  • Table 1: GRACE score should be GRACE risk score Charlson index should be Charlson Comorbidity Index Both should be mentioned or discussed in the Methods section as well   Have these scores been associated with frailty in the literature?

We acknowledge that it should be mentioned that those scores were calculated and the reason why. GRACE score was included in the variables due to its consistent prognostic value in the setting of acute myocardial infarction. As far as we know, it has not been significantly associated with frailty in the literature. On the other hand, comorbidity is frequent in frail patients and has been associated with poor prognosis, independently of frailty; after an acute myocardial infarction (Sanchis MCP 2017). Consequently, Charlson comorbidity index was included in the variables of the study due to its potential predictive role.

The following sentences has been added to methods section:

“GRACE score was calculated in each patient due to its prognostic value in the setting of myocardial infarction [18]. In addition, Charlson Comorbidity Index was included in the collected variables, due to its association with a poor prognosis independently of frailty [19,20].”

Moreover, Table 1 has been changed accordingly.

  • Lines 144-145: in the statement "frailty status as defined", are you referring to binary (frailty versus no frailty) or as a continuous variable?

We thank the reviewer for the comment. We refer to binary variables and frailty is defined by the cut-off point mentioned in the methods section. To facilitate interpretation, the sentence was changed including this clarification (lines 146-147):

“frailty status as defined by either scale (i.e. dichotomic variable) was associated with…”

  • The rationale given for non-significance when scores are combined for multivariate analysis is sample size. If truly due to limited numbers, the number in each group assignment should be described in the Figure (Figure 5) or stated in the results clearly.   

Following your appreciation, Figure 5 was updated including the number of patients in each group:

  • I do not believe that there is sufficient data to prove that combining scales improves prognostication, one can only presume by using both together it would be better clinically per say. The data DO however strongly suggest that one or the other is sufficient for the primary endpoint and probably Fried score better overall when taking into account the secondary endpoint. Therefore, the statements in Line 195/Line 267 should be less demonstrative given the lack of this data.  

We acknowledge that the results of the analysis of the combined value of both scales are not strong enough to draw solid conclusions. Therefore, both statements have been modified accordingly.

Lines 198-199: “Both scales are useful and may be chosen for this prognostic assessment. The combination of frailty defined by FFS and CFS may identify the highest risk patients, although its independent association has not been demonstrated.”

Lines 273-274: “Both scales may be used in this setting, and their combination might provide additional value for risk prediction”

Revisions:  

  • Line 19: Change "Fried" to "Fried's Frailty Scale?" Make this change throughout and abbreviate FFS as a suggestion? Throughout this paper I have seen "Fried score," "Fried's scale," "Fried scale," "Fried phenotype," "Fried" - important to use the same phrase throughout for consistency  

We agree with the reviewer that consistency in nomenclature is more correct and avoids confusion. It has been changed to FFS throughout the manuscript.

  • Line 20: What is "AMI?"  if it is acute myocardial infarction, that abbreviation introduced prior and "from" should change to "with"  

This abbreviation has been removed from the manuscript and we refer to it as myocardial infarction.

  • Line 24-25 is confusing: why include "in 66 patients?"  

In order to clarify, it has been changed to “During the first 800 days 34 deaths and 137 admissions occurred”

  • Line 25-26: should clarify that DAOH decreased with increasing scores

This sentence has been rewritten to clarify the association:

“The number of DAOH decreased significantly with increasing scores of both FFS (p<0.001) and CFS (p=0.049)”.

  • Line 37: add "a syndrome" after "is"  

It has been changed to “Frailty is a syndrome characterized by reduced physiologic reserve…”.

  • Line 38: remove "in those older adults"  

It has been changed to “Its prevalence in European countries ranges…”.

  • Line 42: Change "prefrail" to "pre-frail"  

It has been changed to “38.5% pre-frail”.

  • Line 45: Change "strategy" to "strategies"  

It has been changed to “…evidence-based therapies or invasive strategies”.

  • Line 50: Change "approach" to "approaches"  

It has been changed to “There are two kinds of approaches to evaluate frailty…”.

  • Line 64: Change "prefailty" to "pre-frailty"  

It has been changed to “…patients with pre-frailty or frailty …”.

  • Line 67: Change "could" to "would"  

It has been changed to “…severe concomitant disease that would preclude participation…”.

  • Line 81: Change "personal" to "personnel"  

 It has been changed to “Both assessments were performed by trained personnel”.

  • Line 94: Change "of" to "in"  

It has been changed to “…expressed as ordinal variables by percentage in each category”.

  • Line 131: Change "patients, while..." to "patients, and CFS was >5 in 51 (34%) of patients"  

It has been changed to: “The FFS was ≥3 in 87 (58%) patients, and 51 (34%) had a CFS ≥5”.

  • Line 144: Change "additionally" to "In addition,"  

It has been changed to: “In addition, frailty status as defined by…".

  • Line 193: Change "findings" to "finding" and nd "are" to "is" because only one finding is provided the way the statement is made

It has been changed to: “The main finding of our study is that frailty…”.

  • Line 216: Change "personal" to "personnel"  

It has been changed to: “…particularly when used by inexperienced personnel.”

  • Line 225: Change "reattendance" to "readmission"  

It has been changed to: “need for community services or readmission to ED”.

  • Line 227: Change "predicated" to "predicted"  

It has been changed to: “…both FFS Fried and CFS score significantly predicted mortality at follow-up…”.

  • Line 266: Add "in their respective scales" after "driven by the highest scores."

It has been added as suggested: “This is mainly driven by the highest scores in their respective scales.”

Reviewer 2 Report

I congratulate the authors on this well done study comparing the prognostic value of Fried and Clinical Frailty Scale [CFS] following acute myocardial infarction (AMI). The authors have included 150 patients from the single center randomized trial “Intervention in Frailty Versus Usual Care in Frail Patients After an Acute Myocardial Infarction”. The authors find that frailty as assessed by Fried Score or CFS is independently associated with fewer days alive out of hospital during the first 800 days after AMI, suggesting the importance of using frailty as another tool for further risk stratification after AMI. Overall, the manuscript is well written with minimal language and spelling errors. The data presented supports the conclusion drawn and statistical analysis is done well. The topic of frailty is not novel and it is utilized as an important tool to risk stratify patients prior to several cardiac procedures. In this study, the authors have compared the use of 2 frailty scores in patients presenting with AMI. The authors have also appropriately acknowledged some of the limitations of the study, however, it should be noted that this is a single-center study and, therefore, limited in its generalizability. As the authors have noted, there is also a selection bias in selecting individuals who may not be as frail as the authors excluded the patients who had significant concomitant comorbidities or those who had cognitive impairment. 

Suggesting some minor formatting changes to the figures - for example, in figure 1, please make the font of the numbers shown in the bar graph larger so they are easy to visualize. Remain consistent with using the decimal points throughout the manuscript. 

Author Response

I congratulate the authors on this well done study comparing the prognostic value of Fried and Clinical Frailty Scale [CFS] following acute myocardial infarction (AMI). The authors have included 150 patients from the single center randomized trial “Intervention in Frailty Versus Usual Care in Frail Patients After an Acute Myocardial Infarction”. The authors find that frailty as assessed by Fried Score or CFS is independently associated with fewer days alive out of hospital during the first 800 days after AMI, suggesting the importance of using frailty as another tool for further risk stratification after AMI. Overall, the manuscript is well written with minimal language and spelling errors. The data presented supports the conclusion drawn and statistical analysis is done well. The topic of frailty is not novel and it is utilized as an important tool to risk stratify patients prior to several cardiac procedures. In this study, the authors have compared the use of 2 frailty scores in patients presenting with AMI. The authors have also appropriately acknowledged some of the limitations of the study, however, it should be noted that this is a single-center study and, therefore, limited in its generalizability. As the authors have noted, there is also a selection bias in selecting individuals who may not be as frail as the authors excluded the patients who had significant concomitant comorbidities or those who had cognitive impairment. 

Suggesting some minor formatting changes to the figures - for example, in figure 1, please make the font of the numbers shown in the bar graph larger so they are easy to visualize. Remain consistent with using the decimal points throughout the manuscript.

We fully thank the reviewer for the interest in our manuscript and the kind comments.

Following your suggestion, we have enlarged the numbers in Figure 1.

The manuscript has been further reviewed to use decimal points in a homogeneous way.